# MULAN: MULTIMODAL PROTEIN LANGUAGE MODEL FOR SEQUENCE AND STRUCTURE ENCODING

## ABSTRACT

Most protein language models (PLMs), which produce high-quality protein representations, use only protein sequences during training. However, the known protein structure is crucial in many protein property prediction tasks, so there is a growing interest in incorporating the knowledge about the protein structure into a PLM. Currently, structure-aware PLMs are trained from scratch or introduce a huge parameter overhead for the structure encoder. In this study, we propose MULAN, a MULtimodal PLM for both sequence and ANgle-based structure encoding. MULAN has a pre-trained sequence encoder and an introduced parameter-efficient Structure Adapter, which are then fused and trained together. According to the evaluation on 9 downstream tasks, MULAN models of various sizes show quality improvement compared to both sequence-only ESM2 and structure-aware SaProt as well as comparable performance to Ankh, ESM3, ProstT5, and other PLMs considered in the study. Importantly, unlike other models, MULAN offers a cheap increase in the structural awareness of the protein representations due to finetuning of existing PLMs instead of training from scratch. We perform a detailed analysis of the proposed model and demonstrate its awareness of the protein structure.

## 1 INTRODUCTION

Proteins, as unbranched heteropolymers, play a pivotal role in nearly all biological functions (Finkelstein & Ptitsyn, 2016). Comprising 20 distinct amino acids, the specific sequence of these amino acids determines the complex three-dimensional (3D) structure of the protein (Anfinsen et al., 1961). Subsequently, this 3D configuration governs the protein's function (Finkelstein & Ptitsyn, 2016). Advances in genome sequencing initiated the growth of the number of publicly available protein data, unveiling a vast resource for understanding the molecular basis of life. The application of modern machine learning techniques for a better understanding of protein sequences can boost the development of diverse fields such as drug discovery, protein design, and biotechnology.

The abundance of protein sequences and their text-like nature made it possible to apply top-performing approaches from natural language processing to proteins. It is tempting to expect that protein sequence information alone would be sufficient for large protein language models (PLMs) to infer protein structure and function. Recently, large PLMs, such as ProtTrans (Elnaggar et al., 2021), ESM2 (Lin et al., 2022), and Ankh (Elnaggar et al., 2023) have made remarkable progress in protein representation learning, surpassing previous approaches across various downstream tasks. However, it appears that the representation abilities of sequence-only PLMs are limited and some kind of structural information should be encoded directly into PLM. This limitation is represented by a significantly better performance (Lin et al., 2023) of structure-infused Alphafold (Jumper et al., 2021) compared to sequence-only ESMFold (Lin et al., 2023).

At the same time, structural information about a huge amount of proteins has also become easily available with the revolutionary method AlphaFold (Jumper et al., 2021). Recently, several structural protein language models (SPLMs) were proposed. For example, SaProt (Su et al., 2023) and ESM3 (Hayes et al., 2024) have successfully added some knowledge about the protein structure into the model, showing a better performance compared to sequence-only PLMs. Additionally, existing SPLMs require training from scratch (Wang et al., 2023; Heinzinger et al., 2023; Hayes et al., 2024; Li et al., 2024). Furthermore, hybrid sequence-structure models add a large structure encoder with

a size similar to the base PLM used, resulting in both parameter and computational overhead during training and inference (Chen et al., 2024; Wang et al., 2023; Zhang et al., 2023a; Wang et al., 2022; Li et al., 2024). Such overheads may influence the applicability of such models in high-throughput pipelines, eg. computational drug discovery and protein design.

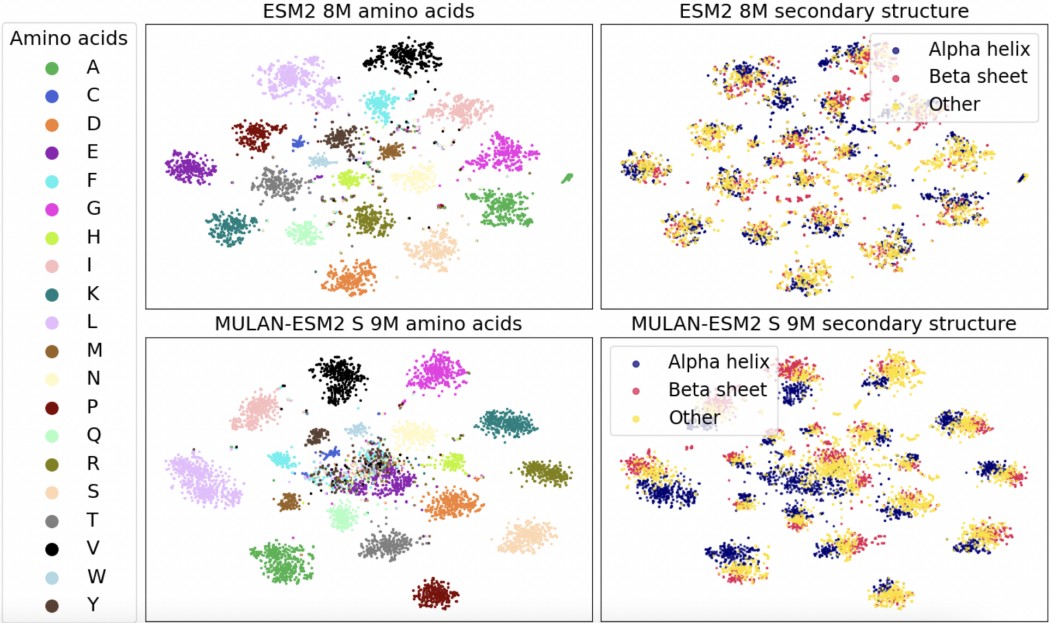

Figure 1: Visualization of residue embeddings of MULAN-ESM2 S 9M and ESM2 8M on CASP12 dataset. We use different colors for amino acid residue types (left) and for the 3 states of secondary structure (right). The details of the experiment are presented in Section 4.3.

In our study we present a simple yet effective structure-aware PLM, that is computationally and parameter-efficient and does not require training from scratch. Our main contributions are:

- We introduce MULAN, a novel MULtimodal PLM for both sequence and ANgle-based structure processing. We propose the Structure Adapter, a lightweight MULAN module that uses residue torsion angles to represent the protein structure. Our model can work on top of existing PLMs through PLM finetuning, so it offers a cheap increase of structural awareness due to avoiding training from scratch.

- We evaluate the obtained structure-aware protein representations on a wide range of downstream tasks and compare MULAN to relevant baselines. We show that adding MULAN to both ESM2 and SaProt of various sizes is beneficial for the quality of the downstream tasks. The main improvements are shown for protein-protein interaction prediction (up to $0.1$ in AUROC for MULAN-ESM2 L), and for the protein GO annotation ($0.023$ for medium and $0.012$ for large MULAN-ESM2 in $F_{max}$ for GO CC).

- We perform an extensive ablation study to highlight the effectiveness and demonstrate the structural awareness of MULAN embeddings (see Section 4.2 and Figure 1).

## 2 METHOD

### 2.1 MULAN ARCHITECTURE

**Structural information** In this study, we propose MULAN, which is a MULtimodal encoder PLM for both sequence and ANgle-based structure processing. MULAN uses the pre-trained base PLM and has the Structure Adapter – a module we introduce to incorporate the knowledge about the 3D protein structure (see Fig. 2a). In our experiments, we use ESM2 architecture, initializing the base PLM from ESM2 or SaProt models. However, MULAN can be based on other PLMs.

We use the information about the protein backbone torsion angles, which are conventionally called $\phi$ and $\psi$ and ensure the protein backbone flexibility. We also use all amino acid residue side chain torsion angles, which are conventionally called $\chi_i$ (up to five $\chi$ angles for amino acid arginine) and provide flexibility of the residues' side chains. Missing $\chi$ angles together with undefined terminal $\phi$ and $\psi$ angles are filled with the reserved padding value, which results in an angle vector $[\phi, \psi, \chi_1, \chi_2, \chi_3, \chi_4, \chi_5]$ for each residue. The residue torsion angles are rotation- and translation-invariant; thus, they are easy to use inside a transformer model.

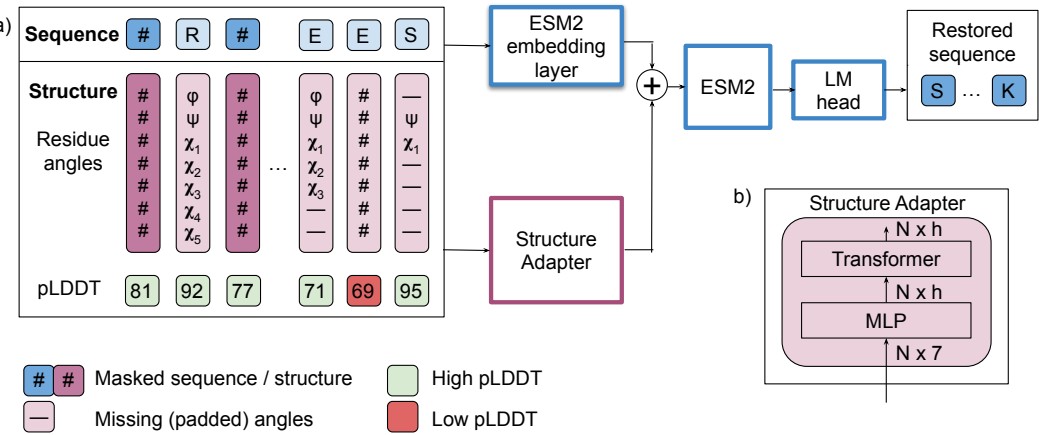

Figure 2: The architecture of MULAN. a) MULAN processes sequence inputs with the ESM2 embeddings module, while structure inputs are passed to the Structure Adapter. Both sequence and structure embeddings are summed up and passed to the ESM2 model, which is then finetuned. Sequence-only ESM2 modules (blue) are initialized from the pre-trained ESM2 checkpoint. Structure processing modules are shown in pink. b) The architecture of the Structure Adapter.

**The Structure Adapter** The proposed Structure Adapter is used to support the multimodality of our model and to fuse structural information with the sequence-only PLM. The Structure Adapter is a small encoder for the protein structure, which consists of an MLP followed by a Transformer layer. The MLP projects the residue angle vector to the residue angle embedding with dimension $h$, while the Transformer layer processes all protein angle embeddings at once (see Fig. 2b). Similarly to ESM2, we use rotary positional embeddings (Su et al., 2024) inside the Transformer layer to encode the order of residue angle vectors. Given a protein of length $N$, the Structure Adapter returns an $N \times h$ angle embedding, or a structure bias. Finally, angle embeddings are added to initial ESM2 residue embeddings of the same dimension $h$ as a structure bias. The resulting structure-aware residue embeddings are then passed through the ESM2 model. Due to the small size of the Structure Adapter, it does not add an overhead to the training or inference time compared to the base PLM.

## 2.2 TRAINING PROCEDURE AND STRUCTURE MASKING

We initialize the base model with a pre-trained ESM2 checkpoint, so during training, we aim to finetune ESM2 using additional structure inputs. We utilize the same masked language modeling objective (MLM) (Devlin et al., 2018) as ESM2 for sequences, randomly masking $15\%$ of tokens in a batch. In this work, we apply a similar masking strategy to the Structure Adapter. Residue angles are masked or replaced jointly with the corresponding residue letters. $80\%$ of the time, angles are masked with a pre-defined mask value, which are then passed to the Structure Adapter. $10\%$ of the time, the angle vector is replaced by a random angle vector from the same protein, while in the rest cases, residue angles remain unchanged. Note, that both reserved padding and masking angle values can have arbitrary values with absolute values higher than $\pi$ in order not to mix with real angle values (we take $4$ and $-4$ in our experiments).

For each residue, AlphaFold produces an estimate of its confidence with a $0 - 100$ scale – predicted local distance difference test score (pLDDT). This measure corresponds to the AlphaFold predicted score on the lDDT-C$\alpha$ metric (Mariani et al., 2013). We found out that only reliable structural information should be passed to the model. We observed that passing low-confidence predictions

into the model worsens its performance (see Section 4.1). Therefore, residue angle vectors with pLDDT less than 70 are considered low confidence, and we mask these structural inputs (see Fig. 2).

# 3 EXPERIMENTS

## 3.1 TRAINING DETAILS

We use protein structures for Swiss-Prot proteins from AlphaFold Protein Structure Database (AlphaFold DB) (Varadi et al., 2022) for the pre-training stage ($503, 724$ structures). We filter out proteins with a length of less than 30 amino acids. The final set comprises $501, 348$ proteins. Following Rives et al. (2021), we randomly selected the validation set of $5000$ proteins.

The small MULAN model (MULAN-ESM2 S 9M) was initialized from the ESM2 8M checkpoint. For the medium-sized models (MULAN M 37M), we use 35M ESM2 or SaProt, while for large models (MULAN L 652M) we take 650M ESM2 or SaProt checkpoints.

During the pre-training stage, we randomly crop proteins that are longer than $1022$ residues to the length of $1022$. We follow Rives et al. (2021) and use dynamic batch size without the concatenation of proteins along the sequence dimension during training. Also, to fully utilize GPU and minimize the amount of padding, we use sorted batching with dynamic batch size as in (Gonzalez et al., 2023): we keep a fixed number of tokens in the batch and form the batch from proteins with similar lengths. The extended training details and hyperparameters are detailed in Appendix A.

## 3.2 DOWNSTREAM TASKS

We tested our model on 10 downstream tasks, of which the first 9 are main tasks and the last one, Secondary structure prediction, served as a sanity check to verify the MULAN's structure awareness. We summarize the information about all these datasets in Appendix B.1 and Table 5. For the prediction of the Localization (both binary and 10-class), Thermostability, Metal Ion Binding, Gene Ontology (GO), and Human Protein-Protein Interaction (HumanPPI) we follow the setup of Su et al. (2023). Fluorescence prediction was taken from Ankh (Elnaggar et al., 2023) setup. We used the Secondary structure prediction task from TAPE (Rao et al., 2019) benchmark.

AlphaFold structures for proteins from the datasets were retrieved from AlphaFold DB by Uniprot accession number if available. PDB IDs were mapped to UniProt accession numbers and retrieved from AlphaFold DB. If no protein identifier was available or there was no AlphaFold model for the UniProt accession number in the database, the protein was modeled by the standalone version of AlphaFold. For all described datasets we keep the original data splits. For the Secondary structure prediction, we use original experimental PDB structures for evaluation. The description of datasets and processing steps is detailed in Appendix B.1.

To evaluate all binary classification tasks we use the area under the ROC curve (AUC); for multiclass classification we measure accuracy; for multilabel GO annotation task, we follow (Gligorijević et al., 2021) and use $F_{max}$ score; and for the regression tasks we measure Spearman's correlation coefficient (SCC).

## 3.3 DOWNSTREAM TASK EVALUATION

**Downstream model** To evaluate the performance of SPLMs, we extract protein embeddings from the last layer of a model. For protein-level tasks, we perform the average pooling of embeddings. Downstream task prediction is done using the model with the Light Attention architecture (Stärk et al., 2021), which was designed to work with protein embeddings and shows better results than an MLP. We detail the architecture of the downstream model in Appendix B.2

**Experimental setup** We train the downstream model and select optimal hyperparameters for each downstream task and embedding model independently based on the metric on the validation set. We set a fixed grid of hyperparameters for all downstream tasks, which includes the dropout rate and intermediate representation sizes of a downstream model. The grid and all used hyperparameters are detailed in Appendix B.3. We use AdamW optimizer ($\beta_1 = 0.9$, $\beta_2 = 0.999$) with a fixed learning

Table 1: Performance gains caused by adding MULAN to ESM2 and SaProt models compared to the initial models used for MULAN (MUL) initialization. The quality is measured on 9 downstream tasks. Positive gains are in bold

| Model name | Localization 10-cl. / binary | Thermo-stability | Fluorescence | Metal Ion | Human PPI | GO CC / MF / BP |
|---|---|---|---|---|---|---|
| | acc $\uparrow$ / AUC $\uparrow$ | SCC $\uparrow$ | SCC $\uparrow$ | AUC $\uparrow$ | AUC $\uparrow$ | $F_{max} \uparrow$ |
| MUL-ESM2 S | **.003**/**.001** | **.014** | **.022** | **.070** | **.007** | **005**/**.059**/**.024** |
| MUL-ESM2 M | **.008**/ − .002 | **.001** | −.006 | −.011 | **.045** | **.023**/**.011**/**.010** |
| MUL-ESM2 L | **.008**/ − .001 | **.013** | **.012** | −.012 | **.098** | **.012**/ − .005/**.0** |
| MUL-SaProt M | −.007/**.004** | **.005** | **.003** | **.017** | **.048** | **.004**/ − .001/**.002** |
| MUL-SaProt L | −.003/ − .002 | −.008 | **.001** | **.026** | **.048** | **.005**/**.005**/**.006** |

rate (in most setups it is $5 \cdot 10^{-5}$). The batch size is equal to $8192$ for all experiments. The model is trained for 200 epochs, and we select the intermediate checkpoint with the best validation metric.

## 4 RESULTS AND DISCUSSION

### 4.1 RESULTS

As baseline models for comparison we take sequence-only PLMs (ProteinBert (Brandes et al., 2022), ESM2 (Lin et al., 2022) and Ankh (Elnaggar et al., 2023)); structure-aware SPLMs (ProstT5 (Heinzinger et al., 2023), SaProt (Su et al., 2023) and ESM3 (Hayes et al., 2024)); hybrid sequence-structure models (S-PLM (Wang et al., 2023) and PST (Chen et al., 2024)), which are top-performing to the best of our knowledge. We do not compare to ESM-GearNet (Zhang et al., 2023a), because SaProt has reported to be better (Su et al., 2023), and LM-GVP (Wang et al., 2022) and GearNet (Zhang et al., 2023b) were surpassed by ESM-GearNet (Zhang et al., 2023a). Another hybrid model DeProt (Li et al., 2024) does not have implementation available. We train MULAN on top of two model families: ESM2 and SaProt AF of various sizes. The quality of the protein representations on the considered downstream tasks is reported in Table 2. Note that we report results for models of different sizes in separate parts of the table and highlight the best results in bold also separately according to the model size. Also, we do not report error bars for the results due to the computationally expensive process of re-training PLMs. Our main findings are discussed below.

**Adding MULAN is beneficial for different PLMs** First of all, we aim to highlight the downstream performance gains introduced by MULAN when applied to both ESM2 and SaProt models. These improvements are shown in Table 1: we report gains of MULAN models compared to the base PLMs used for MULAN initialization. For example, MULAN-ESM2 S refers to the gains of this MULAN model compared to the small ESM-2 8M model. Positive gains are in bold. The results demonstrate that MULAN is effective on both small, medium, and large models. We show that the performance of both considered PLMs was improved in most cases by adding the proposed Structure Adapter.

MULAN demonstrates the most impressive results when applied to the small ESM2 model, significantly increasing the quality of Metal Ion Binding prediction (.070 increase in AUROC) as well as protein function prediction: GO MF (.059 in SCC) and Fluorescence (.022 in SCC). This fact indicates that both protein property prediction and interaction prediction tasks benefit from the structural input. Overall, MULAN shows the best results in HumanPPI (for all base models there is a significant improvement, up to $0.1$ in AUROC for MULAN-ESM2 L). Also, MULAN increases downstream quality in GO CC and GO BP tasks.

Furthermore, we show that MULAN-ESM generally offers higher performance gains compared to MULAN-SaProt. We suppose that it is because SaProt is already a structure-aware model, while ESM2 is sequence-only. Moreover, the results show that MULAN is a general-purpose PLM that can produce high-quality results for protein downstream tasks of different nature. We show that this simple strategy can enhance PLM performance for all considered models. We expect that MULAN

Table 2: Comparison of the performance of various PLMs and SPLMs on 9 main downstream tasks. The table is split into sections according to model sizes. We indicate the best results in bold for each section separately. For large models second best results are underlined

| Model name | Localization 10-cl. / binary | Thermo-stability | Fluore scence | Metal Ion | Human PPI | GO CC / MF / BP |
|---|---|---|---|---|---|---|
| | acc $\uparrow$ / AUC $\uparrow$ | SCC $\uparrow$ | SCC $\uparrow$ | AUC $\uparrow$ | AUC $\uparrow$ | $F_{max} \uparrow$ |
| **Small models** | | | | | | |
| ProteinBert 16M | .692/**.952** | .636 | **.610** | .754 | .687 | .447/.542/.415 |
| ESM2 8M | .729/.948 | .666 | .579 | .731 | .698 | .490/.529/.400 |
| MULAN-ESM2 S | **.732**/.949 | **.680** | .591 | **.801** | **.705** | **.495**/**.588**/**.424** |
| **Medium models** | | | | | | |
| ESM2 35M | .760/**.966** | .689 | .592 | .793 | .751 | .489/.621/.443 |
| MULAN-ESM2 M | **.768**/.964 | .690 | .586 | .782 | **.796** | **.512**/**.632**/**.453** |
| Saprot AF 35M | .767/.960 | .699 | .639 | .783 | .731 | .501/**.632**/.440 |
| MULAN-SaProt M | .760/.964 | **.704** | **.642** | **.800** | .779 | .505/.631/.442 |
| **Large models** | | | | | | |
| Ankh base 450M | .804/.966 | .703 | .630 | .837 | .758 | .510/.686/.495 |
| Ankh large 1.2B | .806/.954 | .668 | .638 | .787 | .738 | .517/**.692**/**.501** |
| ProstT5 1.2B | .773/.954 | .693 | .633 | .805 | .663 | .518/.687/.484 |
| PST 1.1B | .820/.965 | .689 | .618 | .831 | .809 | .526/.676/.475 |
| S-PLM 704M | .796/.950 | .678 | .584 | .767 | .731 | .486/.671/.466 |
| ESM3 1.4B | .751/.951 | .695 | .663 | **.846** | .704 | .512/.673/.471 |
| ESM2 650M | .808/.969 | .694 | .601 | .781 | .754 | .523/.678/.479 |
| MULAN-ESM2 L | .816/.968 | .707 | .613 | .769 | **.852** | .535/.673/.479 |
| SaProt AF 650M | **.846**/**.974** | **.711** | .668 | .776 | .720 | .540/.658/.464 |
| MULAN-SaProt L | .843/.972 | .703 | **.669** | .802 | .768 | **.545**/.663/.470 |

can be applied to any large PLM (for example, Ankh), further improving their performance by a computationally-efficient fine-tuning.

**MULAN further boosts structure-aware SaProt**   Even though SaProt already uses the protein structure information, MULAN-SaProt still generally increases the quality of protein representations for both medium and large models with the highest improvements made for the protein interaction prediction tasks (HumanPPI and Metal Ion Binding). At the same time, for some downstream tasks, the performance was kept at the original level of SaProt, hence, MULAN does not degrade the performance of the original model. These results highlight the fact that the currently used Foldseek structural data may be not enough to fully encode the protein structure. Still, there is room for improvement in the use of 3D structure, and the Structure Adapter has shown success in the further enrichment of protein representations with structural information.

**MULAN works on par with other SPLMs**   Although our main contribution is in the lightweight improvement of existing PLMs, and MULAN does not aim to achieve the best results among all existing approaches for protein representation learning, we still compare our model to all relevant baselines. We report all baseline and MULAN models in Table 2, splitting it according to the model sizes. MULAN-ESM2 S shows the best results compared to small ESM2 8M and even twice bigger ProteinBert 16M model. To the best of our knowledge, in the medium-sized models, there are only already considered ESM2 and Saprot, so we have discussed them earlier.

As for the large models, there is no clear superiority of one model over others among baselines: there are downstream tasks where each model can show good results. This is another reason why MULAN does not aim to be better than every existing model. However, MULAN performs strictly better than S-PLM and is comparable to ProstT5, ESM3, and PST (there are approximately half of the tasks better in MULAN L).

Table 3: Ablation study of the training pipeline for ESM2 8M and MULAN-ESM2 S. The first section corresponds to ESM2 8M results without the Structure Adapter, while the second section – for MULAN-ESM2 S. The best results are shown in bold, second best are underlined

| Model name | Localization 10-cl. / binary | Therm. stab. | Fluore scence | Metal Ion | Human PPI | GO CC / MF / BP |
|---|---|---|---|---|---|---|
| | acc ↑ / AUC ↑ | SCC ↑ | SCC ↑ | AUC ↑ | AUC ↑ | $F_{max}$ ↑ |
| **ESM2 8M** | .729/.948 | .666 | .579 | .731 | .698 | .490/.529/.400 |
| ESM2 + finetune | .715/.946 | .679 | .576 | .743 | .728 | .476/.545/.410 |
| **MULAN-ESM2 S** | **.732**/**.949** | .680 | **.591** | **.801** | .705 | .495/**.588**/**.424** |
| without pLDDT | .712/.937 | **.684** | .582 | .760 | .738 | **.496**/.579/.420 |
| from scratch | .668/.922 | .663 | .550 | .709 | **.765** | .461/.514/.373 |

**MULAN offers cheap structural awareness** Remarkably, MULAN achieves these quality improvements with minimal effort. Firstly, unlike SaProt, ProstT5, and ESM3, it does not require training from scratch. Instead, we perform a lightweight finetuning of a pre-trained PLM for several epochs (up to three days on one GPU). In contrast, SaProt reports that the computational cost of training is similar to ESM-1b (Su et al., 2023), which results in three months of training. Thus, MULAN is very computationally-friendly in terms of training time and applying it to new models. Secondly, MULAN adds the minimal parameter overhead. The proposed Structure Adapter uses only one Transformer layer, which results in 0.3% parameter overhead for MULAN L with 652M parameters. At the same time, other hybrid models add much more parameters: PST adds 69% overhead, and S-PLM adds 13.8% more parameters. These additional large structure encoders significantly decrease the inference speed and require more powerful GPUs, which is undesirable in practical scenarios. However, MULAN works strictly better than S-PLM and there is no clear superiority of MULAN or PST over one another. Thus, we conclude that even a lightweight approach can reach the level of much larger models.

## 4.2 ABLATION STUDY

In the experiments below we perform a detailed analysis of the performance of MULAN-ESM2 S to reduce the amount of computations. The extended results with all ablation studies and analysis of the used hyperparameters are shown in Appendix C. We discuss the importance of sequence and structure modalities, architectural choice, and learning rate strategies there.

**The Structure Adapter is the key factor for improvement** We aim to evaluate the importance of the additional Structure Adapter and the influence of the training procedure we use. For this purpose, we finetune the pure ESM2 8M on our training dataset (ESM2 + finetune experiment) to show that our training dataset and the finetuning procedure itself do not lead to a significant performance boost on downstream tasks. The results of the evaluation demonstrate that finetuning of ESM2 on our data is not enough, do the main contribution to the performance boost is done by the Structure Adapter (see ESM2 vs ESM2 + finetune vs MULAN-ESM2 experiments in Table 3).

**Masking structural inputs with pLDDT for noise reduction** We show that it is useful to mask uncertain residues in AlphaFold structure models (with pLDDT > 70) before passing them to the Structure Adapter: see MULAN vs MULAN without pLDDT masking experiments in Table 3. Most of the downstream tasks benefit from this trick, most likely due to the noise reduction in input angles.

**Starting from the pre-trained model is necessary** We demonstrate the benefits of finetuning the pre-trained ESM2 model instead of training MULAN from scratch. Despite the addition of the structure bias to the initial ESM2 embeddings, it is still possible and useful to adjust all model weights for new inputs and not to lose the base knowledge of a pre-trained model. We applied the same training procedure to the randomly initialized MULAN (see Table 3: MULAN vs MULAN from scratch), and the obtained results are much worse compared even to ESM2 8M, which indicates the need for much more time for training MULAN from scratch.

Table 4: Comparison of the performance of PLMs on the secondary structure prediction task. The table is split into sections based on the model size. The best results for each section are in bold

| Model name | 3-state, accuracy ↑ | | | 8-state, accuracy ↑ | | |
|---|---|---|---|---|---|---|
| | CASP12 | TS115 | CB513 | CASP12 | TS115 | CB513 |
| **Small models** | | | | | | |
| ESM2 8M | .732 | .798 | .765 | .602 | .677 | .623 |
| MULAN-ESM2 S | **.894** | **.918** | **.895** | **.815** | **.854** | **.806** |
| **Medium models** | | | | | | |
| ESM2 35M | .752 | .828 | .809 | .619 | .707 | .669 |
| MULAN-ESM2 M | .886 | .901 | .877 | .789 | .814 | .769 |
| SaProt AF 35M | .900 | .924 | .910 | .805 | .848 | .817 |
| MULAN-SaProt M | **.905** | **.927** | **.912** | **.807** | **.852** | **.818** |

**Architecture ablation**    Following Su et al. (2023); Heinzinger et al. (2023), we try to use Foldseek sequences to represent the structure in the same manner as the Structure Adapter does. Also, we experimented with structure features prediction heads and additional structure-related loss functions. As a result, we tried to add an additional Foldseek embedding layer or Contact and Angle prediction heads. According to our experiments (see Table 7), we did not notice a significant quality improvement compared to the base MULAN induced by these modifications (see Table 7). We explain these architectural modifications in detail and discuss the obtained results in Appendix C.1.

## 4.3 SECONDARY STRUCTURE PREDICTION

We aim to show the awareness and proper use of 3D structure by MULAN. For this purpose, we evaluate our model on the secondary structure prediction downstream task (see Table 4). We report results on CASP12 (Moult et al., 2018), TS115 (Yang et al., 2018) and CB513 (Cuff & Barton, 1999) datasets. Initially, MULAN was trained using AlphaFold structures. It masks uncertain residue predictions based on the AlphaFold pLDDT score. We evaluate the quality of secondary structure prediction using the initial datasets with experimental structures. For such type of structures, pLDDT is not applicable, so we pass angle information for all residues into MULAN without masking.

**Structural awareness of MULAN**    According to the results from Table 4, MULAN models demonstrate the awareness of the protein secondary structure. They surpass similar-sized ESM2 models by a large margin. The same is true for large models, whose results are shown in Table 10 in Appendix D. We do understand that the correct information about the secondary structure can be derived from angle inputs as well as from the Foldseek tokens used by SaProt. Hence, this experiment is done only to demonstrate that MULAN actively uses the 3D structure.

**Visualization**    The quality of the separation of the PLM representations according to some physical or structural property can serve as evidence of the model's physical and structural awareness. We perform a comparison of MULAN-ESM2 S and ESM2 8M representations according to the visual quality of the t-SNE (Van der Maaten & Hinton, 2008) visualization. We plot residue-level embeddings from the last layer of both models on the CASP12 dataset (Moult et al., 2018). We highlight in color different amino acid residue types on the left and different secondary structure types (3 states) on the right (see Fig. 1). On the one hand, MULAN shows much higher awareness of the secondary structure compared to ESM2: for most amino acid clusters three secondary structure types are separated, while for ESM2 they are mostly mixed up. On the other hand, MULAN does not lose the initial knowledge about the amino acid properties gained from the ESM2 model. All these findings are in line with the pre-training strategy. ESM2 has sequences as inputs, so it aligns amino acid representations in separate clusters. MULAN has both sequences and structure inputs; therefore, its representations are well-aligned in both domains.

## 5 RELATED WORK

### 5.1 SEQUENCE-BASED MODELS

Protein sequences are similar to human language: like letters are assembled into words that, in turn, form sentences, amino acids are chained into protein sequences that encode protein 3D structure which determines function. This resemblance makes it promising to apply best practices from the natural language processing field for solving protein-related tasks. Most PLMs are pre-trained with a masked language modeling (MLM) objective (Devlin et al., 2018): a part of the input sequence's residues is randomly masked or replaced with other residues, and then the model aims to predict these corrupted tokens using the remaining sequence context. Models from the Transformer family Heinzinger et al. (2019); Brandes et al. (2022); Xiao et al. (2021); Rives et al. (2021); Lin et al. (2023); Elnaggar et al. (2021; 2023) have made huge progress in the protein representation learning, among which ESM2 (Lin et al., 2023) and Ankh (Elnaggar et al., 2023) are currently showing the best results. They show high performance in various downstream tasks, for example, the prediction of protein secondary structure, residue contacts, sub-cellular localization, and the effect of mutation.

### 5.2 STRUCTURE-INFORMED MODELS

The amino acid sequence solely defines the protein structure (Anfinsen et al., 1961), which, in turn, defines all protein properties, including its function. However, the sequence alone is not sufficient enough for PLMs to infer all information about the protein (Lin et al., 2023). Thus, attempts to infuse PLMs with structural context were made. To improve PLM's capabilities, Zhang et al. (2024) proposed to finetune ESM2 on the remote homology detection task, which seems to implicitly incorporate protein structure-based features into the model. Indeed, the protein structure is more conserved than the protein sequence (Chothia & Lesk, 1986), and homology search, which is a more sensitive task than finding similar protein sequences, gives the model more structural knowledge.

Recently, the idea of using protein 3D structure directly during the model pre-training has been given a lot of attention from the research community. ProstT5 (Heinzinger et al., 2023), the first structure-augmented PLM, uses Foldseek, a special structural alphabet (van Kempen et al., 2023) (3Di) that describes the tertiary structure. As a result, each protein can be represented with either an amino acid sequence or a string of Foldseek letters of the same length that carries information about tertiary interactions. Incorporating that information into PLM was done by finetuning a sequence-only ProtT5 model to translate between the amino acid sequence and 3Di sequence to obtain structure-aware protein representations. Another structure-informed PLM, SaProt (Su et al., 2023), uses the 3Di alphabet to encode the structure similarly to ProstT5. It represents each residue as a combination of amino acid and 3Di letters and is trained with MLM from scratch. This approach gives SaProt an improvement over ESM2 on various downstream tasks. Tan et al. (2024) suggest an adapter-based approach (SES-Adapter) that works with Foldseek sequences and residue secondary structure annotation. However, it is an approach for structure-aware downstream task tuning rather than a general-purpose PLM. It needs to be applied and trained for each downstream task separately and does not provide protein structure-aware embeddings for general use. As a result, SES-Adapter cannot be compared to MULAN. Recently, ESM3 (Hayes et al., 2024) has incorporated sequence, structure, and function modalities for the protein representation learning task as well as protein generation. Similarly to MULAN, they embed and fuse different modalities of the protein, but train a large PLM from scratch.

### 5.3 HYBRID MODELS

The graph-like tertiary structure of proteins gives ideas of infusing PLMs with structure information via Graph Neural Networks. In this setup, pre-training is left as MLM only (Mansoor et al., 2021; Zheng et al., 2023) or is augmented by Masked Structure Modeling task where not only parts of the sequence are masked but parts of the structure too (LM-GVP (Wang et al., 2022), GearNet (Zhang et al., 2023b), MIF (Yang et al., 2023). In ESM-GearNet (Zhang et al., 2023a) it was proposed to fuse the protein sequence and structure information from state-of-the-art PLMs with graph structure encoders (GearNet). The authors used various pre-training strategies including diffusion-based, and reported a performance boost compared to ESM2 and GearNet on several downstream tasks. DeProt (Li et al., 2024) works similarly to ESM-GearNet. DeProt uses local protein structure around

the residue to get the residue-level structure encoding. Li et al. (2024) proposed DeProt, another way of Chen et al. (2024) presented PST, an approach to combine a graph encoder and a PLM and to jointly train them to obtain structure-aware protein representations. PST modifies the self-attention mechanism of the underlying PLM and trains the whole model jointly. The graph encoder has as many parameters as the base PLM used. PST shows better results compared to relevant baselines, eg. ESM2 and ESM-GearNet. Wang et al. (2023) presents S-PLM, a contrastive learning approach to jointly train protein sequence and structure encoders. A structure encoder is a Swin-Transformer that works on residue-residue contact matrices and is trained from scratch. S-PLM does not take protein structure explicitly during inference, relying only on the learned structure-aware representations.

## 6 CONCLUSION

In this paper, we propose MULAN, a novel multimodal 3D structure-aware protein language model for both sequence and structure processing. MULAN works on top of a pre-trained PLM and has the introduced lightweight Structure Adapter that processes residue dihedral angles. Our model fine-tunes the pre-trained PLM model offering a cheap incorporation of the knowledge about the protein structure into the model. Also, unlike other hybrid structure-aware models, the Structure Adapter of MULAN adds minor parameter overhead to the base PLM: 0.3% for large MULAN. We train MULAN models of various sizes and evaluate their protein representations on 9 downstream tasks. For most of the downstream tasks, our model demonstrates an increase in performance compared to ESM2 and SaProt models, which were used for MULAN initialization. The best results were shown for the protein-protein interaction and the protein GO annotation prediction. Additionally, MULAN demonstrates comparable performance to ESM3, ProstT5, and other PLMs considered in the study, while having a faster training or inference pipeline. Finally, we demonstrate the structural awareness of our model in multiple experiments.

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

## A    TRAINING DETAILS

We used AdamW optimizer ($\beta_1 = 0.9$, $\beta_2 = 0.999$) (Loshchilov & Hutter, 2017) for all MULAN models. MULAN-ESM2 S was trained with the learning rate $10^{-4}$ during 20 epochs and with 12000 tokens per batch. The training process is run on 1 Tesla V100 GPU and lasts approximately 18 hours (375k steps).

Both medium and large models were trained using 1 Tesla H100 GPU for approximately 360k steps. For medium models, we used 32000 tokens per batch and the learning rate $10^{-5}$, and it resulted in 65 epochs and 1.5 days. For large we took 10000 tokens per batch and the learning rate $5 \cdot 10^{-6}$, and it resulted in 20 epochs and 3 days.

## B    DOWNSTREAM TASKS

### B.1    DOWNSTREAM DATASETS

We follow Su et al. (2023) and use their setup for protein Localization, Thermostability, Metal Ion Binding, GO, and HumanPPI. Localization prediction from DeepLoc dataset (Almagro Armenteros et al., 2017) has two tasks: classification of proteins into 2 and 10 categories, which both reported. For Thermostability prediction, the "Human-cell" split from FLIP benchmark (Dallago et al., 2021) is used. It relies on human data from Meltom atlas (Jarzab et al., 2020). Also, one of the considered downstream tasks is Metal Ion Binding: we predict whether there are metal ion-binding sites in the protein (Hu et al., 2022). The prediction of protein-protein interaction for human proteins (HumanPPI) (Pan et al., 2010) is taken from PEER benchmark (Xu et al., 2022). We predict GO terms (Gligorijević et al., 2021) and use all three branches independently: Molecular Function (MF), Biological Process (BP), and Cellular Component (CC). GO annotation is a multilabel prediction task. For all listed downstream tasks we use data provided by (Su et al., 2023), so all used AlphaFold protein structures are available in the AlphaFold database.

Fluorescence prediction is done based on the data of the fluorescence intensity of green fluorescent protein (GFP) mutants (Sarkisyan et al., 2016). We follow the setup of Ankh evaluation and use the split from TAPE (Rao et al., 2019) benchmark. We built an AlphaFold structure of the wild-type GFP protein and used Rosetta relaxation protocol (Simons et al., 1999) for the generation of mutant

Table 5: Downstream tasks summary. We use the following abbreviations: mult. class. – multilabel or multiclass classification, bin. class. – binary classification

| Task name | Prediction level | Task type | Evaluation metric | Data split sizes |
|---|---|---|---|---|
| | | | | train / valid / test |
| Localization (10-cl.) | protein | mult. class. | accuracy | $8,743 / 2,190 / 2,745$ |
| Localization (binary) | protein | bin. class. | AUROC | $5,477 / 1,336 / 1,731$ |
| Thermostability | protein | regression | SCC | $5,056 / 639 / 1,336$ |
| Fluorescence | protein | regression | SCC | $21,446 / 5,362 / 27,217$ |
| Metal Ion Binding | protein | bin. class. | AUC | $5,066 / 662 / 665$ |
| HumanPPI | prot. pair | bin. class. | AUC | $26,317 / 234 / 180$ |
| GO CC | protein | mult. class. | $F_{max}$ | $26,225 / 2,904 / 3,350$ |
| GO MF | protein | mult. class. | $F_{max}$ | $26,225 / 2,904 / 3,350$ |
| GO BP | protein | mult. class. | $F_{max}$ | $26,225 / 2,904 / 3,350$ |
| Secondary structure | residue | mult. class. | accuracy | $8,678 / 2,170 / 21; 115; 434$ |

Table 6: Downstream task hyperparameters: learning rate (lr), dropout rate (dropout), intermediate representation sizes $h_1$ and $h_2$

| Task name | lr | dropout | $h_1$ | $h_2$ |
|---|---|---|---|---|
| Localization (10-class) | $5 \cdot 10^{-5}$ | $\{0.1, 0.2\}$ | $\{1536, 1024, 512\}$ | $\{512, 256\}$ |
| Localization (binary) | $5 \cdot 10^{-5}$ | $\{0.1, 0.2\}$ | $\{1536, 1024, 512\}$ | $\{512, 256\}$ |
| Metal Ion Binding | $5 \cdot 10^{-5}$ | $\{0.1, 0.2\}$ | $\{1536, 1024, 512\}$ | $\{512, 256\}$ |
| HumanPPI | $5 \cdot 10^{-6}$ | $0.2$ | $\{1536, 1024, 512\}$ | $\{512, 256\}$ |
| GO CC / MF / BP | $5 \cdot 10^{-4}$ | $0.1$ | $1536$ | $768$ |
| Thermostability | $5 \cdot 10^{-5}$ | $\{0.1, 0.2\}$ | $\{1536, 1024, 512\}$ | $\{512, 256\}$ |
| Fluorescence | $5 \cdot 10^{-5}$ | $\{0.1, 0.2\}$ | $\{1536, 1024, 512\}$ | $\{512, 256\}$ |
| Secondary structure | $1 \cdot 10^{-4}$ | $0.1$ | $1024$ | $512$ |

3D structures. GFP_wt sequence was taken from the original dataset (Sarkisyan et al., 2016). The reference GFP structure was provided to Rosetta to build mutant structures. Since these are single mutants, their structures should not differ a lot from the structure of wild-type GFP, and simple relaxation is enough. Then, we used pLDDT scores from the initial GFP structure for training on all mutant proteins.

Moreover, we evaluate our model on the secondary structure prediction task which is taken from TAPE benchmark (Rao et al., 2019). We report results on three test datasets: CASP12 (Moult et al., 2018), TS115 (Yang et al., 2018) and CB513 (Cuff & Barton, 1999), both for 3-state and 8-state setups. For this task, only experimental structures are available, so we use them as an input to MULAN. For all experimental structures, we pass all residue angles without masking into the MULAN. This is done because of the absence and inapplicability of pLDDT to the experimental structures.

We summarize the information about all downstream tasks in Table 5.

## B.2 DOWNSTREAM MODEL ARCHITECTURE

Downstream task prediction is done using the model with the Light Attention architecture (Stärk et al., 2021), which was designed to work with protein embeddings and shows better results than an MLP. The only difference is that we extend it by adding two extra intermediate layers $L_1$ and $L_2$: $L_i = \text{Dropout}(\text{ReLU}(\text{BatchNorm}(\text{Linear}))) : \mathbb{R}^{h_{i-1}} \to \mathbb{R}^{h_i}$, where $h_1$ and $h_2$ are the model hyperparameters, and $h_0$ is the initial embedding dimension. They are added before the output Linear layer, which projects embeddings of size $h_2$ into the downstream task target dimension.

### B.3 DOWNSTREAM TASK HYPERPARAMETERS

Here we present the grid used to select optimal hyperparameters for all downstream tasks (see 6). For the GO task, we have selected and fixed hyperparameters that perform well for all PLMs. We do not perform grid search because of the long time required for a single evaluation. Moreover, for GO it was optimal to increase the learning rate because of the much bigger output dimension in this task: up to 1943 classes for GO BP. For the HumanPPI task, the optimal learning rate differs from the base one due to the different nature of the task: we need to input two concatenated protein embeddings instead of one to the downstream model. Also, we reduced the grid for HumanPPI (take 0.2 dropout rate) to decrease the number of required computations. The batch size is equal to 8192 for all experiments, and the training time is 200 epochs, but we select the intermediate checkpoint with the best validation metric. Since we use the secondary structure prediction task only to show the structural awareness of the model, we fix hyperparameters for the downstream task evaluation for a faster model evaluation.

## C ABLATION STUDIES

In this section, we present all ablation experiments that were performed during the selection of the architecture and the training procedure of MULAN as well as the hyperparameter tuning.

### C.1 WAYS OF INFUSING THE PROTEIN STRUCTURE INTO MULAN

There are many ways of representing the protein structure and infusing the structural information into the PLM. We have chosen to use the residue torsion angles as a model input. However, we have tried different approaches discussed below.

**Foldseek as another structure input** Following Su et al. (2023); Heinzinger et al. (2023), we try to use Foldseek sequences to represent the structure. We try to do it in the same manner as the Structure Adapter does. We add the Foldseek embedding layer, which is then finetuned together with ESM2. Also, we tried to combine both the Structure Adapter and the Foldseek embedding layer. The extended architecture of MULAN is shown in Figure 3.

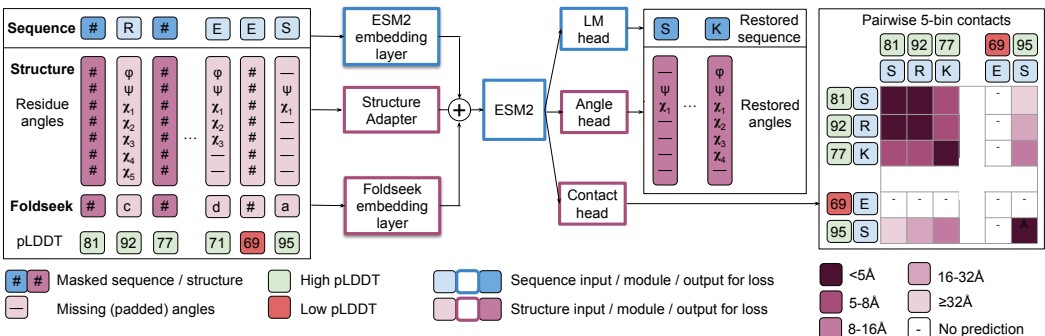

Figure 3: The extended MULAN architecture. There is an extra Foldseek embedding layer and two output structure features prediction heads: Contact and Angle heads. Both sequence, structure, and Foldseek embeddings are summed up and passed to the ESM2 model, which is then finetuned. Sequence-only ESM2 modules are initialized from the pre-trained ESM2 checkpoint and are shown in blue. Structure processing modules are shown in pink.

**Coordinates as another structure input** Additionally, we try to use xyz coordinates of $C_\alpha$ atoms to represent the structure. We treat them with another Structure Adapter that projects a 3-dimensional coordinate vector into an embedding of size $h$. Then, the resulting coordinate embedding is summed up with ESM embeddings and angle embeddings.

**The objective function** MULAN is trained in the same way as ESM2 and most other PLMs: it uses the masked language modeling objective (MLM). One can use the knowledge about the protein

Table 7: Ablation study of the training pipeline for ESM2 8M and MULAN-ESM2 S. The first section corresponds to ESM2 8M results without the Structure Adapter, while the second section – for MULAN-ESM2 S. The best results for each section are shown in bold, the second best are underlined

| Model name | Localization 10-cl. / binary | Therm. stab. | Fluore scence | Metal Ion | Human PPI | GO CC / MF / BP |
|---|---|---|---|---|---|---|
| | acc ↑ / AUC ↑ | SCC ↑ | SCC ↑ | AUC ↑ | AUC ↑ | $F_{max}$ ↑ |
| **ESM2 8M** | .729/.948 | .666 | .579 | .731 | .698 | .490/.529/.400 |
| ESM2 + Angle | .709/.942 | .665 | .562 | .735 | .740 | .481/.540/.410 |
| ESM2 + Contact | .706/.942 | .675 | .553 | .703 | .692 | .477/.533/.406 |
| **MULAN-ESM2 S** | .732/**.949** | .680 | .591 | **.801** | .705 | .495/.588/.424 |
| MULAN + Angle | .728/.943 | .655 | .589 | .751 | .727 | .480/.561/.404 |
| MULAN + Contact | .722/.939 | **.686** | **.599** | .738 | .656 | .489/.592/.428 |
| MULAN + Foldseek | .730/.942 | .633 | .571 | **.801** | .736 | .479/.584/.430 |
| MULAN + Coordinates | .712/.947 | .684 | .588 | .797 | **.755** | .487/.585/.420 |
| ESM2 + Foldseek | **.735**/.947 | .664 | .564 | **.801** | .733 | **.498**/**.593**/**.431** |

structure not only as model input but also in the additional loss functions. We have experimented with two extra heads for the prediction of protein structure features, which were added to MULAN to increase its structural awareness. Firstly, we tried to restore masked angle inputs similarly to the original sequence MLM objective. This is done using the Angle Prediction Head, which has the same architecture as the ESM2 language modeling (LM) head except for the output dimension, which is 7, the number of residue angles. We use mean squared error (MSE) as a loss function.

Moreover, we use binarized pairwise distances between all residues in the form of a distance matrix as another source of structural data. For this purpose, we introduce a Contact Prediction Head. We binarize distances into 5 bins separated by the following distances: {5Å, 8Å, 16Å, 32Å}. Further, we predict $N \times N$ contact matrix based on the $N_{heads} * N_{layers} \times N \times N$ model attention weight tensor from all $N_{layers}$ layers with $N_{heads}$ attention heads. We use cross entropy (CE) loss and compute it only for residues with confident AlphaFold predictions to avoid training on noisy data. To sum up, we use the following objective function $\mathcal{L}$ during training:

$$\mathcal{L} = \mathcal{L}_{CE}(\text{sequence}) + \alpha \mathcal{L}_{MSE}(\text{angles}) + \beta \mathcal{L}_{CE}(\text{contacts}), \tag{1}$$

where $\alpha$ and $\beta$ are floating point training hyperparameters. However, in the experiments that use structure features prediction heads, we utilize $\alpha = 5$ and $\beta = 0.5$ as they have shown the best performance on downstream tasks.

Similarly to the structural inputs, we compute both contact and angle prediction losses only on residues with the reliable AlphaFold structure with pLDDT not less than 70.

**Architecture ablation study results**   We have conducted an ablation study to identify the most important modules among those we have discussed earlier. We consider the Structure Adapter and Foldseek embedding layer as input modules that encode structure for MULAN. Also, we experiment with the Angle and Contact prediction heads as output modules that add structural knowledge to MULAN embeddings via the structure-related loss functions. In all experiments, we follow the same training procedure as with MULAN.

Firstly, we conduct loss-only experiments without structure inputs. For this purpose, we finetune ESM2 8M with additional structure features prediction heads (ESM2 + Angle / ESM2 + Contact). The results of the evaluation are presented in Table 7 in the first section with ESM2. According to the results, using the structure information in the loss function is not sufficient for improvement on downstream tasks: only metrics for HumanPPI increased significantly.

Secondly, we evaluate MULAN with the extra structure features prediction heads (MULAN + Angle / MULAN + Contact). Moreover, we experiment with the Foldseek embedding layer as another source of structure inputs that can be passed to our model. We try both a combination of the Structure

Table 8: Contribution of different modalities in MULAN-ESM2 S performance

| Model name | Localization 10-cl. / binary | Therm. stab. | Fluore scence | Metal Ion | Human PPI | GO CC / MF / BP |
|---|---|---|---|---|---|---|
| | acc ↑ / AUC ↑ | SCC ↑ | SCC ↑ | AUC ↑ | AUC ↑ | $F_{max}$ ↑ |
| **MULAN-ESM2 S** | .732/.949 | .680 | .591 | .801 | .705 | .495/.588/.424 |
| sequence-only | .698/.945 | .628 | .564 | .700 | .777 | .430/.363/.329 |
| structure-only | .332/.546 | .397 | .100 | .487 | .569 | .334/.177/.261 |

Table 9: Comparison of the performance of MULAN-ESM2 S with different learning rate (lr) strategies. If two learning rates are reported, the lowest corresponds to ESM2 modules, while the highest – for the Structure Adapter. For each section of the table best results are shown in bold

| Model name | Localization 10-cl. / binary | Thermo- stability | Fluore scence | Metal Ion | Human PPI | GO CC / MF / BP |
|---|---|---|---|---|---|---|
| | acc ↑ / AUC ↑ | SCC ↑ | SCC ↑ | AUC ↑ | AUC ↑ | $F_{max}$ ↑ |
| lr $5 \cdot 10^{-4}/5 \cdot 10^{-5}$ | .716/.936 | .674 | .581 | .743 | **.732** | .493/.555/.407 |
| lr $1 \cdot 10^{-3}/1 \cdot 10^{-4}$ | .727/.944 | .675 | .575 | .785 | .680 | **.496**/.578/.423 |
| lr $1 \cdot 10^{-4}$ for all | **.732/.949** | **.680** | **.591** | **.801** | .705 | .495/**.588/.424** |

Adapter with a Foldseek embedding (MULAN + Foldseek) and the Foldseek embedding layer alone (ESM2 + Foldseek). The results are shown in the second part of Table 7. According to them, there is no clear evidence of superiority of one approach over another, we did not notice a consistent improvement. Thus, we decided to keep MULAN architecture as simple as possible and not to use additional heads. Also, we keep only the Structure Adapter as an input structure processing module. We do it because the use of the Foldseek embedding layer reduces the quality of Thermostability and Fluorescence prediction.

## C.2 CONTRIBUTION OF SEQUENCE AND STRUCTURE MODALITIES

Since MULAN has access to both sequence and structure angle data, we aim to analyze the contribution of both modalities. We show MULAN results with completely masked structural information (MULAN sequence-only experiment) and with the completely masked sequences (MULAN structure-only experiment) in Table 8. The results show that input protein structure angles highly influence the quality of MULAN embeddings, resulting in even lower quality for a sequence-only MULAN compared to the initial ESM2 8M model. The same is true for the structure-only scenario: the structure alone is not enough for good protein representations. This experiment shows the importance of both modalities and the use of the structural information by our model.

## C.3 LEARNING RATE STRATEGIES

We used the same learning rate for all parameters of MULAN during training. However, we initialize the whole protein encoder from the ESM2 pre-trained checkpoint, while the Structure Adapter is newly initialized. This fact suggests the possibility of using a smaller learning rate for ESM2 modules compared to the Structure Adapter in order not to harm the pre-trained weights a lot. This idea was suggested and shown its effectiveness in Zhang et al. (2023a), where they have a similar combination of randomly initialized and pre-trained modules. We follow the suggested setup and decrease the learning rate for ESM2 modules by a factor of 10. According to the results, for MU-LAN there is no clear benefit of using the reduced learning rate for ESM2 modules (see Table 9), so we decided to keep the learning rate constant for all MULAN modules to reduce the number of used hyperparameters.

Table 10: Comparison of the performance of PLMs on the secondary structure prediction task. The table is split into sections based on the model size. The best results for each section are in bold. For large models second best results are underlined

| Model name | 3-state, accuracy ↑ | | | 8-state, accuracy ↑ | | |
|---|---|---|---|---|---|---|
| | CASP12 | TS115 | CB513 | CASP12 | TS115 | CB513 |
| **Small models** | | | | | | |
| ESM2 8M | .732 | .798 | .765 | .602 | .677 | .623 |
| MULAN-ESM2 S | **.894** | **.918** | **.895** | **.815** | **.854** | **.806** |
| **Medium models** | | | | | | |
| ESM2 35M | .752 | .828 | .809 | .619 | .707 | .669 |
| MULAN-ESM2 M | .886 | .901 | .877 | .789 | .814 | .769 |
| SaProt AF 35M | .900 | .924 | .910 | .805 | .848 | .817 |
| MULAN-SaProt M | **.905** | **.927** | **.912** | **.807** | **.852** | **.818** |
| **Large models** | | | | | | |
| ProstT5 1.2B | .858 | .887 | .891 | .747 | .800 | .797 |
| PST 1.1B | .853 | .893 | .880 | .749 | .802 | .773 |
| ESM3 1.4B | **.949** | **.969** | **.955** | **.921** | **.948** | **.923** |
| ESM2 650M | .821 | .871 | .867 | .706 | .772 | .751 |
| MULAN-ESM2 L | .867 | .908 | .890 | .778 | .833 | .792 |
| SaProt AF 650M | .926 | .948 | .945 | .844 | .897 | .869 |
| MULAN-SaProt L | .922 | .949 | .935 | .841 | .860 | .870 |

## C.4 ADDITIONAL EXPERIMENTS

**Addition vs concatenation of embeddings**   MULAN uses the structure bias from the Structure Adapter in a manner of positional embeddings: these structural embeddings are added to the main amino acid embeddings from the ESM2 model. However, one may concatenate these embeddings instead of summing up. This approach leads to different objectives used for amino acid embeddings (MLM) and structure embeddings (MLM for angle restoration). Also, concatenation leads to an increase in the length of the content passed to the Transformer model, causing significant memory and time overheads. The results of the experiment with the concatenation of embeddings did not show any benefit compared to the base setup.

**Importance of angle masking**   In our pre-training strategy, both amino acids and corresponding angle vectors are masked together. However, there are two other options that we have tested: independent masking of angles and residue letters and no angle masking at all. These experiments have shown worse results than the base approach. We explain it with the fact that residue letters and their angle vectors are connected. For example, if the letter is masked, and the angle vector has no side chain torsion angles defined, then the range of possible outcomes decreases from all 20 amino acids to only two: Glycine and Alanine. The opposite also holds: the known residue letter helps to restore the corresponding residue angle vector or at least the number of residue angles. Thus, we keep the joint masking strategy to force MULAN to learn as much information as possible.

## D    STRUCTURAL AWARENESS OF MULAN

According to the results from Table 10, all MULAN models demonstrate the awareness of the protein secondary structure. They surpass similar-sized ESM2 models by a large margin. Moreover, even a small MULAN utilizes the knowledge about the protein structure better than structure-aware ProstT5 and PST with more than a billion parameters. We do understand that the correct information about the secondary structure can be derived from angle inputs as well as from the Foldseek tokens used by SaProt or direct secondary structure types used by ESM3. Hence, this experiment is done only to demonstrate that MULAN actively uses the 3D structure.

Table 11: Comparison of the computational resources required for various large PLMs and SPLMs

| Model name | # of params | Inference time, ms | VRAM, MiB | No training from scratch |
|---|---|---|---|---|
| ESM2 | 650M | 132 | 3274 | - |
| SaProt AF | 650M | 126 | 3274 | ✗ |
| MULAN-ESM2 L | 652M | 134 | 3368 | ✓ |
| S-PLM | 704M | 103 | 3288 | ✓ |
| PST | 1.1B | 283 | 6922 | ✓ |
| Ankh large | 1.2B | 236 | 5292 | - |
| ProstT5 (half) | 1.2B | 107 | 2974 | ✗ |
| ProstT5 (full) | 1.2B | 217 | 5258 | ✗ |
| ESM3 | 1.4B | 355 | 13948 | ✗ |

# E    COMPUTATIONAL REQUIREMENTS COMPARISON

We perform an analysis of the time and memory requirements for different large PLMs and SPLMs. We measure the inference time required for one forward pass as well as the amount of required VRAM on the Nvidia Tesla V100 GPU with 16Gb of VRAM. We take approximately the longest possible protein that can be handled on this GPU by ESM3: UniProt protein Q07009 with 702 residues, which is present in the test set of the GO annotation downstream task. We run 10 model inference runs and measure the average inference time for the protein.

The results are shown in Table 11. According to the results, MULAN offers the best combination of quality and efficiency.

- Having the same computational costs as S-PLM, MULAN gives significantly better downstream results.
- MULAN requires 2 times less time and memory compared to PST, having similar downstream quality.
- MULAN requires only finetuning without training from scratch as in SaProt, ProstT5, and ESM3, having similar downstream quality.

