# OpenReview forum: "MULAN: Multimodal Protein Language Model for Sequence and Structure Encoding"
_ICLR.cc/2025/Conference — Submitted to ICLR 2025_

### Official Review · Reviewer_EUj2 · 2024-10-22

**Soundness:** 2
**Presentation:** 3
**Contribution:** 2
**Rating:** 5
**Confidence:** 5

**Summary:**

The authors introduced a new method called MULAN to enhance existing protein language models with angle-based structural information. By introducing a lightweight Structure Adapter module, they incorporated structural information efficiently. The authors evaluated their mothod on a wide range of tasks with comparison to relevant baselines. The experiments illustrate the effectiveness of MULAN. They also did ablation studies to highlight the structural awareness of MULAN embeddings.

**Strengths:**

The authors did many experiments to demonstrate the advantages of MULAN. They included many relevant works and made careful comparison among these methods. The details of experiments for pre-training and downstream fine-tuning are illustrated clearly, and the analysis of experimental results are comprehensive. Also they did many ablation studies to investigate the effectiveness of their method.

**Weaknesses:**

**1. Motivation**

The authors mentioned that existing sequence-structure models tend to be large that may hinders their applicability in downstream tasks. Although MULAN only introduces a lightweight structural module, it still may not be easy to use depend on the size of backbone PLM, e.g. ESM-2 650M or 3B. The proposed method seems not to be advantageous from the prospective of the motivation.

**2. Lack of novelty.**

 There are many related works regarding protein sequence and structure co-modeling, as listed by the authors. The way of adding structural bias into per-residue embeddings has been introduced many times, such as ESM-3 and DeProt. The structural features are only torsion angles which are commonly used such as Foldseek. Also, the pre-training data derives from the Swiss-Prot database, which has been widely used and I suspect its diversity for both sequence and structure modality.

**3. About some claims**

Through extensive experiments, the authors stated that **MULAN is beneficial for different PLMs** and **The Structure Adapter is the key factor for improvement**. Although they did an ablation study showing that further pre-training ESM-2 on Swiss-Prot resulted in worse performance than incorporating MULAN, I would suspect whether it works for structure-based PLM, such Saprot. MULAN didn't get outstanding performance compared to other structure-based Models, and the features it used were very simple, even less than Foldseek which uses both torsion angles and spatial distances. Could the authors try pre-training Saprot without MULAN on their training set and see how the downstream performance would be?


**4. Paper writing.**

In part 2.1 MULAN Architecture, the authors introduced the structure features they used. However, I think they should include some basic introductions about what are torsion angles $\phi$ and $\psi$. Only including the name of angles is confusing and may result in misunderstanding about used features.

**Questions:**

Please see the questions proposed above.

---

> ### Author Response · Authors · 2024-11-19
>
> Dear Reviewer EUj2,
>
> Thank you for your suggestions and questions, we address them below.
>
> > Motivation. The authors mentioned that existing sequence-structure models tend to be large that may hinders their applicability in downstream tasks. Although MULAN only introduces a lightweight structural module, it still may not be easy to use depend on the size of backbone PLM, e.g. ESM-2 650M or 3B. The proposed method seems not to be advantageous from the prospective of the motivation.
>
> Our motivation is to have a model that
> - requires minimal changes to the base architecture and can directly leverage pre-trained PLM checkpoints;
> - makes a lightweight add-on -- the Structure Adapter -- to enable structural inputs, which leads to the same inference time and memory requirements;
> - works on par with existing structure-aware PLMs that are larger and slower or require training from scratch.
>
> To strengthen our point we have measured the inference time and GPU memory requirements of large PLMs needed for the same protein (see Appendix E and Table 11). In short, MULAN offers the best trade-off between quality and efficiency:
> - Having the same computational costs as S-PLM, MULAN gives significantly better downstream results;
> - MULAN requires 2 times less time and memory compared to PST, having similar downstream quality;
> - MULAN requires only finetuning without training from scratch as in SaProt, ProstT5, and ESM3, having similar downstream quality.
>
> > Lack of novelty. There are many related works regarding protein sequence and structure co-modeling, as listed by the authors. The way of adding structural bias into per-residue embeddings has been introduced many times, such as ESM-3 and DeProt. The structural features are only torsion angles which are commonly used such as Foldseek. Also, the pre-training data derives from the Swiss-Prot database, which has been widely used and I suspect its diversity for both sequence and structure modality.
>
> The novelty of our work is in line with our motivation written above. We show that MULAN has simple architecture and is effective both in terms of downstream quality and computational resources needed to run the model.
>
> > MULAN didn't get outstanding performance compared to other structure-based Models, and the features it used were very simple, even less than Foldseek which uses both torsion angles and spatial distances.
>
> Indeed, we use very basic structural features, namely, the torsion angles. However, our investigation shows that torsion angles are enough to restore the protein structure with a small error, while Foldseek is not enough to decode the structure. Thus, torsion angles can be more expressive than Foldseek in our setup.
>
> Ablation experiments with the use of Foldseek inputs, Foldseek inputs together with torsion angles, and residue-residue distance matrices as a prediction objective were outlined in our paper, Appendix C.1 and Table 7. These experiments have shown no clear benefits so we decided to keep the architecture simple yet effective. Also, we have done additional ablation experiment with passing 3D coordinates of residue C-alpha atoms as an input to MULAN. These coordinates are encoded similarly to torsion angle vectors with another Structure Adapter, that projects 3 coordinates to an embedding for the model. We made a sum of ESM, angle, and coordinate embeddings. We added this experiment and its results to Appendix C.1. The results show no consistent improvement over the base MULAN version (see Table 7). Overall, residue torsion angles are expressive enough to encode the protein structure.

---

> ### Author Response · Authors · 2024-11-19
>
> > Although they did an ablation study showing that further pre-training ESM-2 on Swiss-Prot resulted in worse performance than incorporating MULAN, I would suspect whether it works for structure-based PLM, such as SaProt. MULAN didn't get outstanding performance compared to other structure-based Models, and the features it used were very simple, even less than Foldseek which uses both torsion angles and spatial distances. Could the authors try pre-training Saprot without MULAN on their training set and see how the downstream performance would be?
>
> We did our ablation studies based on the sequence-only model, particularly ESM2 for two reasons:
> - ESM2 has a small computationally efficient model, and one can experiment a lot with this base model;
> - MULAN is effective for sequence-only PLMs.
>
> Following your suggestion, we additionally fine-tuned the SaProt 35M model on our dataset without the Structure Adapter. We got a similar performance with and without MULAN, our reason is that SaProt already uses the structure (Foldseek instead of torsion angles). Still, the main focus of MULAN is enhancing sequence-only models with structural information, as we already highlighted in the paper (lines 266-268,  “Furthermore, we show that MULAN-ESM generally offers higher performance gains compared to MULAN-SaProt. We suppose that it is because SaProt is already a structure-aware model, while ESM2 is sequence-only.”).
>
> > Paper writing. In part 2.1 of MULAN Architecture, the authors introduced the structure features they used. However, I think they should include some basic introductions about what are torsion angles ϕ and ψ. Only including the name of angles is confusing and may result in misunderstanding about used features.
>
> Thank you for suggesting the revision, we have extended Section 2.1 with additional details.

---

> > ### Comment · Reviewer_EUj2 · 2024-11-20
> >
> > Dear authors,
> >
> > Thank you for trying to address my concerns. However, some questions still exist:
> >
> > **1. About the motivation**
> >
> > I don't understand why you need to add additional lightweight structure adapter on existing sequence-based PLMs rather than making use of open-sourced structure-based PLMs. On the one hand, the inference speed and GPU memory usage are only limited by the backbone models that are used, e.g. 35M or 650M (according to Table 11, I don't see a better performance compared to SaProt or S-PLM). On the other hand, I doubt the gain of perfomance on downstream tasks derived from further pre-training on Swiss-Prot database, not the MULAN module. That's why I suggested you train SaProt on the same dataset and see how it performs. I think you should showcase the result on Table 3 to prove the effectiveness of MULAN.
> >
> > **2. About the residue torsion angles**
> >
> > I cannot agree to that point that **"residue torsion angles are expressive enough to encode the protein structure"**. Even in your Table 7 I could see ESM2 + Foldseek performs better than MULAN + ESM. Furthermore, I think there should be many papers demonstrating the importance of interatomic distances when encoding protein structures. For instance, ProteinMPNN showed that interatomic distances evidently provide a better inductive bias to capture interactions between residues than dihedral angles or N-Ca-C frame orientations, and using protein dihedral angles as node input features did not result in better performance. Therefore, I think only encoding torsion angles is not enough for better modeling protein structure.

---

> > > ### Author Response · Authors · 2024-11-26
> > >
> > > Dear Reviewer EUj2,
> > >
> > > Thank you for the detailed comments on our response.
> > >
> > > > I don't understand why you need to add additional lightweight structure adapter on existing sequence-based PLMs rather than making use of open-sourced structure-based PLMs.
> > >
> > > Using MULAN can be beneficial for adding new protein modalities to existing pre-trained PLMs. Now we are focused on the protein structure, which is already used, for example, in SaProt. However, one can add different modalities, eg. functional annotation or other additional protein information, in the same manner as MULAN. These new modalities can further enrich existing models like SaProt without retraining from scratch.
> > >
> > > > About the residue torsion angles
> > >
> > > Indeed, the expressive power of different kinds of structure representation is quite similar and interchangeable. Our choice to use angles is based on our experimental results (Table 7).
> > >
> > > Best regards

---

> > > > ### Comment · Reviewer_EUj2 · 2024-11-26
> > > >
> > > > Dear authors,
> > > >
> > > > Thanks for the comments. I agree that MULAN could be useful to integrate other modalities to enrich the representation capabilities of existing models. However that requires efforts for new feature encoding and experimental investigations and it goes beyond  the scope of this paper. I think I will keep the score still but thank you again for your patient explanations.
> > > >
> > > > Best regards

---

### Official Review · Reviewer_tgHX · 2024-10-29

**Soundness:** 3
**Presentation:** 2
**Contribution:** 2
**Rating:** 3
**Confidence:** 4

**Summary:**

The authors of this study have identified an interesting gap in the growing PLM space, an approach to enrich PLM-based protein representations with 3D structural information. This is in contrast to existing models, such as ProstT5, ESM3, etc., that initially train on sequence and structural (and other modal) data. They do this in their model MULAN by incorporating a structural adapter module to pre-trained ESM2 models and fine-tuning using a subset SwissProt Alpha Fold predicted structures. They evaluate their protein representations in ten previously described residue- and protein-level tasks. Overall the performance of their model was comparable with existing sequence-only and structurally-aware models that they compared against.

**Strengths:**

The paper presents an interesting and direct approach to incorporate structure information to enhance pre-trained PLMs. The text is generally clear. The technical dimension is robust with the training of multiple models each with hyper-parameter optimization by grid search. The inclusion of an ablation experiment is important and good practice.

**Weaknesses:**

The paper builds on existing technology but is ultimately not novel. The paper itself cites multiple other models that incorporate structure in training or fine-tuning. The encoding of structure with amino acid torsion angles is not new (see https://academic.oup.com/bioinformatics/article/37/2/162/5892762 as one example). If the ultimate utility is in quicker training and inference, the authors should provide quantitative comparison and better motivate why this use-case is significant. Additionally, the performance of the model is mostly on par or slightly worse than existing models (Table 3), which the authors acknowledge but does diminish the significance. The differences in performance are also relatively small.

The authors mention ProstT5 and ESM3 models in Section 6.2, which both include structural information in training. I think it is necessary to compare MULAN with these models to evaluate how well the their approach compares to these models. That way users can compare the performance and cost tradeoffs between the two approaches.

Ultimately, the work is interesting and a nice extension of pLM work but the study is not novel in model design, training set curation or featurization, or evaluation tasks, with at best marginal performance improvement in a fraction of comparisons. It is therefore my recommendation to reject the manuscript.

**Questions:**

1) What fraction of residues are unmasked once the pLDDT threshold is applied?
2) Why is training limited to the Swiss-Prot fraction of Alpha Fold DB? There are millions of predicted structures available, which might boost performance of MULAN?
3) Can you explain some more about the tasks chosen for evaluation? Why these tasks?
4) Why does MULAN perform worse with more parameters (Table 1)? [I found the presentation of this table with bold vs. not bold to be confusing.]
5) Have you considered that some of the downstream task proteins are present in the set that you use to Swiss-Prot set to fine-tune for MULAN? If there is overlap I think the fraction of proteins in each task present in the training set should be reported (could be in Table 5). It is also worth reporting if any of the CASP12, TS115, or CB513 structures are present in the training set. It could be that they are mutually exclusive, but this should be made explicit none the less.

---

> ### Author Response · Authors · 2024-11-19
>
> Dear Reviewer tgHX,
>
> Thank you for the detailed review. We address your questions below.
>
> > If the ultimate utility is in quicker training and inference, the authors should provide quantitative comparison and better motivate why this use-case is significant.
>
> We added the measurements of the amount of GPU memory and time required for the inference of different PLMs on the same protein (see Appendix E and Table 11). In short, MULAN offers the best trade-off between quality and efficiency.
> - Having the same computational costs as S-PLM, MULAN gives significantly better downstream results.
> - MULAN requires 2 times less time and memory compared to PST, having similar downstream quality.
> - MULAN requires only finetuning without training from scratch as in SaProt, ProstT5, and ESM3, having similar downstream quality.
>
> > The authors mention ProstT5 and ESM3 models in Section 6.2, which both include structural information in training. I think it is necessary to compare MULAN with these models to evaluate how well their approach compares to these models. That way users can compare the performance and cost tradeoffs between the two approaches.
>
> Comparisons with ProstT5 and ESM3 were presented in our paper in Table 2. There is also a comparison to other relevant structure-aware PLMs: S-PLM, PST, and SaProt (Table 2). Moreover, there are results measuring the structural awareness of these models in Appendix D (Table 10).
>
> > What fraction of residues are unmasked once the pLDDT threshold is applied?
>
> In our paper, residues are not unmasked according to the pLDDT threshold. We have the following algorithm for masking:
> - Mask 15% of residues for further MLM decoding. This is done together for sequence and the corresponding angle inputs;
> - Additionally, mask only residue angle vectors that have low pLDDT, while keeping the corresponding sequence letters unmasked.
>
> As a result, we still have 15% of sequences masked and used in the training MLM objective. And we have slightly more structure inputs masked. In the dataset there are 15.4% of the residues with pLDDT lower than 70, which results in 28.1% of structure inputs masked on average during training and 15.4% during inference (when MLM is not used).
>
> > Why is training limited to the Swiss-Prot fraction of Alpha Fold DB? There are millions of predicted structures available, which might boost performance of MULAN?
>
> It is enough to have a subset of the Swiss-Prot dataset for MULAN model. However, we experimented with larger datasets. For example, we took the dataset with 17M protein structures that were used in the ProstT5 paper and obtained lower scores compared to training on Swiss-Prot. We concluded that the Swiss-Prot subset contains protein structures of higher quality. Additionally, MULAN requires only a short finetuning stage, so there is no need for the large training dataset.
>
> > Can you explain some more about the tasks chosen for evaluation? Why these tasks?
>
> Most of the datasets in our paper were taken from the SaProt evaluation setup. Also, we extended the downstream tasks with 2 tasks:
> Fluorescence prediction task, which dataset contains mutants of green fluorescent protein. The Fluorescence dataset can be used to measure sensitivity to the mutations of the protein;
> Secondary structure prediction task to check the structural awareness of MULAN.
>
> > Why does MULAN perform worse with more parameters (Table 1)? [I found the presentation of this table with bold vs. not bold to be confusing.]
>
> Thank you, we have clarified the table by adding minus signs explicitly. Concerning the scale of our model, MULAN performs better with more parameters. In the paper, we reported the performance gain to the base model X, when MULAN is added to the X (row MULAN-X). Positive numbers (bold in Table 1) indicate that adding MULAN to X is beneficial, while negative numbers (not bold in Table 1) show that it is not. Table 1 shows that MULAN improves over base models X in most downstream tasks. So, larger MULAN performs better than medium MULAN, which in turn performs better than small MULAN (Table 2). However, the effect of adding MULAN is higher for small models than for large ones.

---

> ### Author Response · Authors · 2024-11-19
>
> > Have you considered that some of the downstream task proteins are present in the set that you use to Swiss-Prot set to fine-tune for MULAN? If there is overlap I think the fraction of proteins in each task present in the training set should be reported (could be in Table 5). It is also worth reporting if any of the CASP12, TS115, or CB513 structures are present in the training set. It could be that they are mutually exclusive, but this should be made explicit none the less.
>
> The finetuning of MULAN is done in an unsupervised manner: we do not use any of the labels from the downstream tasks. Thus, there is no data leakage when using the same proteins during the finetuning of MULAN and further evaluation of the downstream task performance.
>
> In general it is a common practice to use all available data for unsupervised training: modern LLMs are pre-trained on the whole internet, and modern PLMs are pre-trained on all available protein sequences. For example, ESM2 is trained on 250M protein sequences from the UniParc database, which contains proteins from the downstream datasets.
>
> Regarding the secondary structure prediction, the above answer is still valid. There is no data leakage in the secondary structure labels, because MULAN takes angle vectors explicitly as inputs.

---

> ### Author Response · Authors · 2024-11-28
>
> Dear Reviewer tgHX,
>
> As the end of the discussion period is approaching, we would greatly appreciate your feedback on our rebuttals. Please let us know if you have some additional questions we can address to improve our paper and your score.
>
> Best regards

---

### Official Review · Reviewer_5GUe · 2024-11-01

**Soundness:** 3
**Presentation:** 3
**Contribution:** 2
**Rating:** 5
**Confidence:** 4

**Summary:**

This paper presents MULAN, an approach for enhancing protein language models (PLMs) with 3D structural awareness. Built on ESM-2, a sequence-only PLM, MULAN incorporates a specialized "Structure Adapter" module that processes protein dihedral angles. This allows MULAN to combine sequence and structure information efficiently, offering a substantial improvement in predictive performance across multiple downstream tasks, such as protein-protein interaction and molecular function prediction. MULAN’s structure adapter reduces training overhead by enabling the fine-tuning of pre-trained models without extensive structural pre-training. Experimental results show that MULAN consistently outperforms both ESM-2 and SaProt, a structure-aware PLM, indicating the effectiveness of the proposed structural integration.

**Strengths:**

1. MULAN features a straightforward design by incorporating a Structure Adapter module to effectively integrate protein sequence and structure information. This approach requires minimal changes to the base architecture and can directly leverage pre-trained ESM-2 checkpoints, significantly reducing training costs while achieving multimodal fusion.
2. MULAN is thoroughly evaluated across various downstream tasks, demonstrating consistent performance gains over baseline models. The ablation studies rigorously test critical components, such as the Structure Adapter and pLDDT masking, providing clear insights into the model’s effectiveness.

**Weaknesses:**

1. MULAN relies primarily on dihedral angle information for structural inputs, which, while useful, may not fully capture the complexity of protein structure. The integration of additional structural features, such as inter-residue distance matrices or atomic coordinates, or functionally relevant properties like solvent-accessible surface area (SASA), could potentially improve model performance on downstream tasks by providing a more comprehensive structural context.
2. The paper does not include comparisons with the latest structure-aware protein language models, such as ESM-3, which incorporate advanced structural representations. Comparing MULAN with these models would provide a clearer perspective on its relative advantages and limitations within the current landscape of structure-aware PLMs.

**Questions:**

1. Have the authors considered adding other structural features, like inter-residue distances, atomic coordinates, or functional information? Adding these could potentially improve the model’s structural understanding and task performance.
2. How were structural features combined with ESM’s sequence embeddings? Were any experiments conducted to compare these methods?
3. Has the model been tested for multi-task learning, handling tasks like stability prediction, function annotation, and interaction prediction simultaneously? Given the modular Structure Adapter, it would be interesting to see if MULAN can perform multiple tasks at once and how it compares to single-task models.

---

> ### Author Response · Authors · 2024-11-19
>
> Dear Reviewer 5GUe,
>
> We appreciate your suggestions and address your questions below.
>
> > The paper does not include comparisons with the latest structure-aware protein language models, such as ESM-3, which incorporate advanced structural representations. Comparing MULAN with these models would provide a clearer perspective on its relative advantages and limitations within the current landscape of structure-aware PLMs.
>
> Thank you, we would like to point out that all necessary comparisons were done in our paper. ESM3 results, as well as S-PLM, PST, SaProt, and ProstT5 results, are shown in Table 2. There are also results measuring the structural awareness of these models in Appendix D (Table 10).
>
> > Have the authors considered adding other structural features, like inter-residue distances, atomic coordinates, or functional information? Adding these could potentially improve the model’s structural understanding and task performance.
>
> Yes, we have considered and used the following structural features.
> 1. Foldseek sequences were used both together with the angle information and separately.
> 2. Inter-residue distances were used as a prediction objective (Contact Head), where we predict binarized distances between residues.
> 3. Masked angle restoration objective (Angle Head) is used as an additional loss function.
> 4. XYZ Coordinates of C-alpha atoms were also added as another structure input using the Structure Adapter (this new experiment is added in Appendix C.1).
>
> However, there is no significant improvement from these experiments compared to using only torsion angles, so we decided to keep the simple architecture. These experiments are presented in Appendix C.1 and Table 7.
>
> > How were structural features combined with ESM’s sequence embeddings? Were any experiments conducted to compare these methods?
>
> The structural embeddings produced by the Structure Adapter are summed up to the sequence embeddings produced by the ESM embedding layer. This embedding sum is then passed to the ESM2 Transformer layers. We detail it in Section 2.1.
>
> > Has the model been tested for multi-task learning, handling tasks like stability prediction, function annotation, and interaction prediction simultaneously? Given the modular Structure Adapter, it would be interesting to see if MULAN can perform multiple tasks at once and how it compares to single-task models.
>
> MULAN, as most PLMs, eg. ESM2 and SaProt, is trained in an unsupervised manner with a masked language modeling objective. Downstream task labels are not used as a training objective. Protein embeddings from frozen PLM can be extracted after the unsupervised pre-training is done. Then a small downstream model can be trained for a particular task. Thus, multitask learning is generally not the purpose of training PLMs and is out of the scope of our paper.

---

> > ### Comment · Reviewer_5GUe · 2024-11-21
> > **Feedback to author response**
> >
> > Thank you for your detailed response to my comments and for providing additional information. Here is my feedback on your reply:
> >
> > Regarding the integration of structural features:
> > Thank you for elaborating on the experiments involving XYZ coordinates, contact distances, and other structural features, as well as the corresponding results. I noticed that these additional features did not significantly improve model performance, leading to the decision to retain simple angle-based inputs. Have you further investigated why these additional features did not yield the expected improvements? For example, the results of ESM-2+FoldSeek appear to be very close to those of MULAN+ESM2 S. Does this indicate that specific combinations of features (e.g., angle information with FoldSeek) are already sufficiently robust in certain scenarios?
> >
> > Additionally, have you considered integrating multiple features (e.g., XYZ coordinates, angle information, contact distances) into a single model for joint optimization? This could potentially capture more comprehensive structural information. At the same time, the comparison between MULAN and other structure-based models (e.g., more advanced graph-based or hybrid approaches) seems relatively limited in the paper. Could you further explore performance differences with these models and their possible causes? This might provide a more complete perspective on the strengths and limitations of MULAN.

---

> > > ### Author Response · Authors · 2024-11-26
> > >
> > > Dear Reviewer 5GUe,
> > >
> > > Thank you for the detailed comments on our response.
> > >
> > > > Have you further investigated why these additional features did not yield the expected improvements? For example, the results of ESM-2+FoldSeek appear to be very close to those of MULAN+ESM2 S. Does this indicate that specific combinations of features (e.g., angle information with FoldSeek) are already sufficiently robust in certain scenarios?
> > >
> > > Indeed, Foldseek uses torsion angles in its work, so our angle vectors are connected to Foldseek. Our results show that these features are interchangeable and robust.
> > >
> > > > Additionally, have you considered integrating multiple features (e.g., XYZ coordinates, angle information, contact distances) into a single model for joint optimization?
> > >
> > > Sure, we have done it in our paper. Table 7 shows these results. We have explored the combinations anf angle vectors (MULAN) with Angle recovery head, Contact prediction head, Foldseek embeddings and Coordinate Structure Adapter. These experiments are called MULAN + X, where X is the type of additional structure information used. According to the results, the combination of any 2 types of structural data inputs did not increase the downstream quality significantly, so we did not see the strong evidence to go deeper in testing combinations of more types of structural inputs together.
> > >
> > > Best regards

---

### Official Review · Reviewer_3ooa · 2024-11-05

**Soundness:** 3
**Presentation:** 3
**Contribution:** 3
**Rating:** 5
**Confidence:** 4

**Summary:**

This paper proposes MULAN, which fusing protein 3D structure information with 1D sequence information through a structure adapter. In MULAN, the protein 3D structure information is represented as angle vector of backbone and side-chain torsion angles. Then, the structure adapter employs the transformer layer to encode the structure information and fuse it with the 1D sequence information extracted from a pre-trained PLM (Protein Language Model). The structure adapter and the pre-trained PLM is finetune together. Experimental results are performed to evaluate the effectiveness of MULAN.

**Strengths:**

1.	As demonstrated in the paper, MULAN is computationally efficient. It only takes about 3 days for finetuning a large ESM2 with 650 M parameters.
2.	The experimental results on 10 downstream tasks demonstrate the effectiveness of MULAN, especially with relatively sample PLMs.

**Weaknesses:**

1.	The effectiveness of MULAN on large PLM is limited. To achieve best performance in the practical tasks, we usually need to employ the large PLM rather than small PLM. So, improving the effect of MULAN on large PLM is important.
2.	The overall framework of MULAN is quite similar to existing methods, such as ESM-GearNet and LM-GVP. The only difference may be representation of encoding of structure information. Previous employ geometric-aware graph neural networks to encode the protein 3D structure, while MULAN employs a transformer model.
3.	More analysis should be conducted to analysis the advantages of employing the transformer model for encoding structure information rather than geometric-aware graph neural network.

**Questions:**

1.	The expression of line 480 is a little confusing.
2.	In current framework of MULAN, the PLM models are full finetuned. What about applying PEFT (Parameter-Efficient Fine-Tuning) methods to PLMs?

---

> ### Author Response · Authors · 2024-11-19
>
> Dear Reviewer 3ooa,
>
> Thank you for the valuable feedback and for pointing out our strengths! We will address your questions below.
>
> > More analysis should be conducted to analysis the advantages of employing the transformer model for encoding structure information rather than geometric-aware graph neural network.
>
> We focus on Transformer-based models, because they are currently state-of-the-art model in a lot of tasks, while Graph DL models are worse at scale than Transformer models due to the over-smoothing problem of Graph Neural Networks [1].
>
> [1] Chen, D., Lin, Y., Li, W., Li, P., Zhou, J., & Sun, X. (2020, April). Measuring and relieving the over-smoothing problem for graph neural networks from the topological view. In Proceedings of the AAAI conference on artificial intelligence (Vol. 34, No. 04, pp. 3438-3445).
>
> > The expression of line 480 is a little confusing.
>
> Thank you for noticing, we have checked and rewritten this sentence (line 485).
>
> > In current framework of MULAN, the PLM models are full finetuned. What about applying PEFT (Parameter-Efficient Fine-Tuning) methods to PLMs?
>
> Yes, we have conducted the experiments with fine-tuning ESM-2 650M with LoRA. However, this approach gave lower quality in the downstream tasks, so we decided to keep the full finetuning.

---

> > ### Comment · Reviewer_3ooa · 2024-11-25
> > **Thanks for the rebuttal**
> >
> > Thanks for the response. However, I am not convinced about the current response.
> > For example, the GDL models are also going forward with strong performance, this is not a case that the authors do not analysis.
> > I still look forward to the results of PEFT and also the settings you conduct.

---

### Author Response · Authors · 2024-11-19

Dear Reviewers,

Thank you for acknowledging the ablation studies conducted and MULAN's computational effectiveness. We have addressed your comments and valuable suggestions. Your review helped us develop the following revisions of the paper.
1. We have rewritten the sentence on line 485 for clarity, according to the suggestion of Reviewer 3ooa.
2. We have done additional ablation experiment in response to the concerns of Reviewers 5GUe and EUj2 about the expressiveness of torsion angle vectors (Appendix C.1, “Coordinates as another structure input”).
3. We have added a basic introduction about torsion angles $\phi$ and $\psi$ in Section 2.1, lines 108-111, according to the suggestion of the Reviewer EUj2.
4. We have applied the necessary modifications to the Table 1, its caption and the text that refers to this table in Section 4.1 without changing numbers. We added the minus sign explicitly, as suggested by the Reviewer tgHX.
5. We have conducted a quantitative comparison of GPU time and memory required for the inference of different PLMs on the same protein (Appendix E and Table 11), as suggested by the Reviewer tgHX. According to them, MULAN offers the best trade-off between quality and efficiency.

We address each Reviewer’s questions and concerns in detail below.

---

### Meta-Review · Area_Chair_8qDS · 2024-12-20

**Metareview:**

The paper introduces a framework for equipping a sequence-based protein language model with 3D structure awareness by incorporating a structure adapter processing protein dihedral angles.  Overall, the paper is well written. The approach for incorporating 3D structure awareness is  simple and computationally efficient. It's benefits are convincingly demonstrated on multiple downstream tasks.

However, as pointed out by several reviewers, the novelty of the approach is limited given prior work on the same topic (e.g. ESM-GearNet and LM-GVP). Also as recognized by the authors, the benefits of the approach is much greater for smaller models, for which it might then be a better alternative to go ahead with an existing open-source structure-based PLM.

Furthermore, the use of torsion angles might be effective for the downstream tasks considered, but it is questionable that that may be expressive enough to get a meaningful performance boost for other tasks.

To add significant novelty to the submission, it would be desirable to go beyond structure awareness (as suggested by reviewer EUj2), and experiment with incorporating multiple modalities beyond the protein structure (as the later has been done in many prior work), such as those suggested by the authors and study the benefits.  It might also be worth studying various other ways of fusion beyond summation of embeddings. Another suggestion, which is worth investigating would be to conduct  a deeper comparison of transformer-based vs GNN-based structure awareness as noted by reviewer 3ooa, beyond a high level statement that transformers models are state-of-the-art model in a lot of tasks.

**Additional Comments On Reviewer Discussion:**

The authors have provided several clarifying points in response to the reviewers' questions, and have improved their manuscript to provide valuable additional information such as experiments on C_alpha coordinates. However these are not enough to improve the novelty of the methodology.  The reviewers believe more work is needed such as thorough comparison against graph-based models as suggested by reviewre 3ooa, and/or incorporation of other modalities  (see remarks from reviewer EUj2).

---

### Decision · Program_Chairs · 2025-01-22

Reject